# The N-terminal activation function AF-1 domain of ERα interacts directly with the C-terminal AF-2-holding ligand-binding domain to recruit the coactivator proteins

Xiaohui Liu[1,2]*, Yutaka Matsuyama[1], Makiko Sugiyama[1], Keitaro Suyama[1,3], Takeru Nose[1,3], Miki Shimohigashi[4,5], Yasuyuki Shimohigashi[1,5]*

1 Laboratory of Structure-Function Biochemistry, Department of Chemistry, Faculty of Science, Kyushu University, Fukuoka, Japan, 2 Department of Biotechnology and Life Sciences, Faculty of Biotechnology and Life Sciences, Sojo University, Kumamoto, Japan, 3 Faculty of Arts and Science, Kyushu University, Fukuoka, Japan, 4 Division of Biology, Department of Earth System of Science, Faculty of Science, Fukuoka University, Fukuoka, Japan, 5 Risk Science Research Institute, Fukuoka, Japan

* xiaohui@bio.sojo-u.ac.jp (XL); shimo@kyudai.jp (YS)

**Data Availability Statement:** All relevant data are within the manuscript.

## Abstract

Cryoelectron microscopy (cryo-EM) clarified the quaternary structure of the DNA complex of coactivator-bound estrogen receptor alpha (ERα), revealing the adjacency of the N-terminal domain (NTD) and C-terminal ligand-binding domain (LBD). ERα-NTD and LBD constitute activation function 1 (AF-1) and activation function 2 (AF-2), respectively. These domains are essential for transcription activation. Their spatial proximity was judged to be essential for ERα to recruit the SRC coactivator proteins. In the present study, we first evaluated untethered free ERα-NTD(AF-1) [residues 1–180] and its-truncated desNTD(AF-1)-ERα [residues 181–595] in a luciferase reporter gene assay. ERα-NTD(AF-1) was completely inactive, whereas desNTD(AF-1)-ERα exhibited 66% activity of wild-type ERα. Surprisingly, ERα-NTD(AF-1) was found to inhibit desNTD(AF-1)-ERα markedly. Therefore, assuming that ERα-NTD(AF-1) must also inhibit wild-type full-length ERα, we co-expressed ERα-NTD(AF-1) and full-length ERα. As expected, ERα-NTD(AF-1) inhibited ERα in a dose-dependent manner, but non-competitively for 17β-estradiol. When their intracellular transport was examined immunocytochemically, ERα-NTD(AF-1) showed a distinct translocation from the cytoplasm to the nucleus, despite being expressed solely in the cytoplasm without full-length ERα. This nuclear translocation was attributable to a direct interaction between ERα-NTD(AF-1) and full-length ERα consisting of the nuclear localization signal. The present results demonstrated that, in full-length ERα, the N-terminally tethered NTD(AF-1) domain collaborates with the C-terminal LBD(AF-2) for coactivator recruitment.

## Introduction

Estrogen receptor alpha (ERα) is a nuclear receptor (NR) that pleiotropically regulates various physiological processes, including reproduction, development, embryogenesis, and

**Funding:** This work was supported by the Japan Society for the Promotion of Science (JSPS) KAKENHI; grant nos. 25740024, 15K00557, 19K12340 and 22K12395 (to X.L.) and 22221005 and 15H01741 (to Y.S.). This work was also supported in part by a Health and Labour Sciences Research Grant for Research on the Risk of Chemical Substances from the Ministry of Health, Labor, and Welfare of Japan, under the contract/grant no. H20-Chemistry-General-003 (to Y.S.).

**Competing interests:** The authors declare that no competing interests exist.

**Abbreviations:** 3D, three-dimensional; AF, activation function; AI, artificial intelligence; cryo-EM, cryoelectron microscopy; DBD, DNA-binding domain; desNTD(AF-1)-ERα, ERα C-terminal fragment with amino acid residues 181–595; E2, 17β-estradiol; ERα-NTD(AF-1), ERα N-terminal fragment with amino acid residues 1–180; ERα, estrogen receptor alpha; ERE, estrogen response element; H12, α-helix No. 12; LBD, ligand-binding domain; LBP, ligand-binding pocket; NID, nuclear receptor interaction domain; NLS, nuclear localization signal; NR, nuclear receptor; NTD, N-terminal domain; PBS, phosphate buffered saline; PDB, protein data bank; SRC, steroid receptor coactivator.

metabolism. It is activated by the sex hormone 17β-estradiol (E2) [1–4]. ERα consists of 595 amino acid residues and features a unique domain structure, similar to those of NRs, encompassing the N-terminal domain (NTD: amino acid residues 1–180), the DNA-binding domain (DBD: 181–263), and the ligand-binding domain (LBD: 303–595 including the F-domain) (Fig 1A). There is a flexible hinge linker region (*hinge*: 264–302) between DBD and LBD, and these domains are designated alphabetically A/B, C, D, E, and F arranged from the N-terminus to the C-terminus (Fig 1A). E2 binds to a specific site in the ligand-binding pocket (LBP) located within the LBD, which is in a dimeric structure (Fig 1B).

In the agonist-bound structure, α-helix No. 12 (H12) in LBD binds to the cavity of the pocket LBP, forming a lid over the top of the pocket. As a result of this binding, H12 creates the hydrophobic coactivator-binding site on the molecular surface. This is the reason why H12 is called activation function 2 (AF-2). The p160 family transcriptional coactivator proteins (SRC-1, SRC-2, and SRC-3, where SRC means steroid receptor coactivator) interact with H12/AF-2 via their LXXLL amino acid motifs [6–8]. This direct interaction between H12/AF-2 and the SRC coactivator protein is involved in regulating almost all aspects of gene expression [9, 10]. However, when antagonists such as 4-hydroxytamoxifen and raloxifene bind to ERα, their bulky side chains block H12 sterically from an agonist position [11].

In contrast to the structural or functional role of AF-2, that of activation function 1 (AF-1), which is another name for the domain NTD referred to as NTD(AF-1) hereafter (Fig 1A), has never been elucidated in detail. Using cryoelectron microscopy (cryo-EM), Yi et al. (2015) determined for the first time the quaternary structure of an active complex of DNA-bound ERα [5]. On a DNA molecule, the complex was found to consist of two molecules of ERα, two molecules of SRC-3, and one molecule of the secondary coactivator p300 (Fig 1C). The dimeric structure of ERα-LBD confirms the composition of this complex (Fig 1B). Through density map analysis using an ERα-NTD(AF-1)-specific monoclonal antibody, they concluded that the N-terminal ERα-NTD(AF-1) domain is adjacent to ERα-LBD holding AF-2, called LBD (AF-2) hereafter (Fig 1C). The close proximity of NTD(AF-1) and LBD(AF-2) was judged to be a structure essential for recruiting the SRC-3 coactivator protein [5].

In the primary structure of full-length ERα, N-terminal NTD(AF-1) and C-terminal LBD (AF-2) are apart from each other (Fig 1A). The tethering structure that NTD(AF-1) is linked to the central DBD domain must be crucial for the formation of a binding site for SRC-3. In the present study, to evaluate the intrinsic significance of positioning AF-1 at the ERα N-terminus, we tested a synthetic untethered free ERα-NTD(AF-1) for its-truncated desNTD(AF-1)-ERα and full-length ERα. Here we discuss the molecular mechanisms underlying the function of ERα-NTD(AF-1) as a transcriptional activation factor.

## Materials and methods

### Construction of ERα, ERα-NTD(AF-1), and desNTD(AF-1)-ERα plasmids

The expression plasmid full-length ERα/pcDNA3.1 of wild-type full-length ERα was generated by PCR using an ERα cDNA clone (OriGene Technologies, Rockville, MD, USA) followed by cloning into the vector pcDNA3.1(+) (Invitrogen, Carlsbad, CA, USA), as reported [12, 13]. Using full-length ERα as a template, expression plasmids ERα-NTD(AF-1)/pcDNA3.1 and desNTD(AF-1)-ERα/pcDNA3.1 were generated by PCR followed by cloning into pcDNA3.1 (+).

For the immunocytochemical study, the NTD(AF-1) domain was modified at the N-terminus to be tagged with DYKDDDDK, a FLAG-epitope peptide [14], yielding the plasmid FLAG-ERα-NTD(AF-1)/pcDNA3.1. The expression plasmid full-length ERβ/pcDNA3.1 of wild-type ERβ was generated by PCR using an ERβ cDNA clone. Using this as a template,

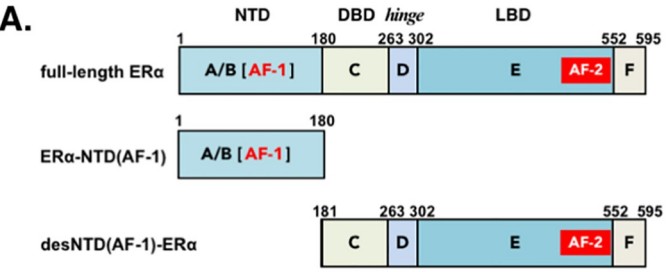

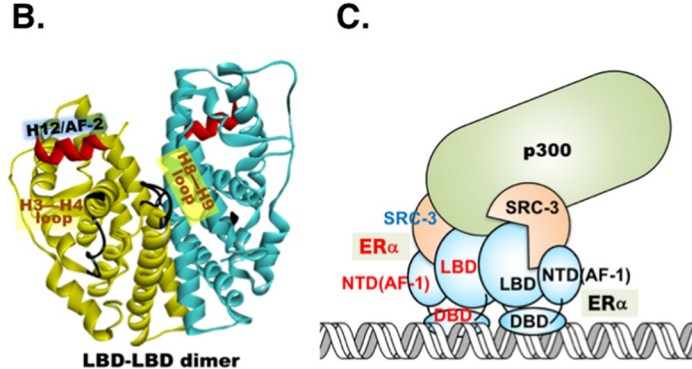

**Fig 1. Constructive features of human estrogen receptor α (ERα). (A)** Schematic bar chart representation of domain structures A–F of ERα, ERα-NTD(AF-1), and desNTD(AF-1)-ERα. Each domain plays a different structural/functional role. Alias names representing domain functions are N-terminal domain, also known as activation function 1 (AF-1) [NTD(AF-1)]; DNA-binding domain (DBD); and ligand-binding domain (LBD). The *hinge* region between DBD and LBD is part of the flexible linker. **(B)** Ordinary 3D schematic ribbon diagram representation of ERα-LBD in the C2 symmetric structure. α-Helix No. 12 (H12) (red) functions as AF-2. The original 3D structure of this dimeric representation is 1G50 (PDB identification cord), which was further analyzed using the software package DiscoveryStudio. **(C)** A symbolized cartoon illustration of the structure of the ERα-coactivator-DNA complex. For this complex, an annotated cryo-EM density map and its illustration were reported by Yi et al. (2015) [5]. SRC-3 is a p160 family transcriptional coactivator protein, and p300 is a secondary coactivator.

FLAG-ERβ-NTD(AF-1)/pcDNA3.1 was prepared similarly. All plasmid constructs were confirmed by DNA sequencing.

## Cell culture and luciferase reporter gene assay

HeLa cells (RCB0007; RIKEN BRC, Tsukuba, Japan) were seeded at $5 \times 10^5$ cells/well of a 6-well microplate for 24 h at 37°C in a 5% $CO_2$ atmosphere, as reported [15–18], using modified Eagle's Minimum Essential Medium (Nissui, Tokyo, Japan) in the presence of 10% (v/v) charcoal-treated fetal bovine serum. The following three plasmids were co-transfected in the total 2.0 mL medium: (i) firefly luciferase reporter plasmid pGL3 (Promega, Madison, WI, USA) containing a triple-repeat estrogen response element (3 × ERE/pGL3; ERE = AGGTCAnnnTGACCT), (ii) expression plasmid full-length ERα/pcDNA3.1, and (iii) pGL4.74 (Promega) internal control plasmid containing the *Renilla* luciferase cDNA [19]. The transfection was performed with Lipofectamine$^{TM}$ 3000 transfection reagent (Thermo Fisher Scientific, Waltham, MA, USA) according to the manufacturer's protocol. The transfected cells were cultured for approximately 24 h at 37°C in a 5% $CO_2$ atmosphere.

For the luciferase reporter gene assay, the cells were harvested using 0.25% trypsin/0.02% EDTA solution and then plated onto 96-well microplates at $5 \times 10^4$ cells/well. After the agonist

E2 was administered at different concentrations, the cells were incubated for another 24 h. The firefly/*Renilla* luciferase activities were measured and normalized using a dual-luciferase reporter assay system (Promega). Light emission was measured on a Fluoroskan FL microplate reader (Thermo Fisher Scientific). Each assay was performed in duplicate at least three times.

To evaluate the transcription activity of desNTD(AF-1)-ER$\alpha$, expression plasmid desNTD (AF-1)-ER$\alpha$/pcDNA3.1 was used instead of full-length ER$\alpha$/pcDNA3.1. To assess the effect of ER$\alpha$-NTD(AF-1) on desNTD(AF-1)-ER$\alpha$, the expression plasmids ER$\alpha$-NTD(AF-1)/pcDNA3.1 and desNTD(AF-1)-ER$\alpha$/pcDNA3.1 were co-transfected into the cells simultaneously at the ratio of X: 1, where X [ER$\alpha$-NTD(AF-1)] = 1, 2, 3, 4, 5, and 6, with the fixed net amount of desNTD(AF-1)-ER$\alpha$ = 250 ng/well. The luciferase reporter gene assays were carried out exactly as described above.

## Co-expression of ER$\alpha$-NTD(AF-1) and full-length ER$\alpha$ or ER$\beta$

**Gene transfection.** An untethered free ER$\alpha$-NTD(AF-1) fragment (1–180) and ER$\alpha$-NTD(AF-1) or epitope-modified FLAG-ER$\alpha$-NTD(AF-1) were tested for their ability to affect the transcription activity of E2 for full-length ER$\alpha$ (1–595). Test solutions were prepared using 6-well microplates, delivering HeLa cells ($5 \times 10^5$ cells) into each well. Expression plasmid ER$\alpha$-NTD(AF-1)/pcDNA3.1 or FLAG-ER$\alpha$-NTD(AF-1)/pcDNA3.1 was first transfected, administered in a dose-dependent manner (0.01–2.00 $\mu$g/well) (Table 1). The total amount of the expression vector and plasmids was maintained at 2.50 $\mu$g/well to utilize the Lipofectamine$^{TM}$ 3000 transfection reagent (Thermo Fisher Scientific), and this was adjusted to 0.25–2.25 $\mu$g/well of the pcDNA3.1 empty vector for each transfection (Table 1). Other expression plasmids were used in fixed amounts: full-length ER$\alpha$/pcDNA3.1 (0.05 $\mu$g/well), 3 × ERE/pGL3 (0.15 $\mu$g/well), and pGL4.74 (0.05 $\mu$g/well) (Table 1).

**Luciferase reporter gene assay.** Approximately 24 h after the gene transfection, cells were harvested and plated onto 96-well microplates at $5 \times 10^4$ cells/well. The transcription activity at each dose was assessed for three E2 concentrations (0.1, 1, and 50 nM).

**Assay of ER$\beta$-NTD(AF-1) for ER$\alpha$.** To evaluate the specificity of ER$\alpha$-NTD(AF-1) for full-length ER$\alpha$, co-expression tests were carried out using the plasmids containing full-length

**Table 1. Preparation of test solutions for ER$\alpha$-NTD(AF-1) inhibition assay: Amount of introduced plasmid genes for a single well of the 6-well cell culture microplate.**

| Plasmids | Amount of each gene introduced (µg/well[1]) | | | | | | | | | | | | | | | |
|---|---|---|---|---|---|---|---|---|---|---|---|---|---|---|---|---|
| | I | II | III | IV | V | VI | VII | VIII | IX | X | XI | XII | XIII | XIV | XV[2] | XVI[3] |
| ER$\alpha$-full[4]/pcDNA3.1 | 0.05 | 0.05 | 0.05 | 0.05 | 0.05 | 0.05 | 0.05 | 0.05 | 0.05 | 0.05 | 0.05 | 0.05 | 0.05 | 0.05 | 0.00 | 0.00 |
| 3 × ERE/pGL3 | 0.15 | 0.15 | 0.15 | 0.15 | 0.15 | 0.15 | 0.15 | 0.15 | 0.15 | 0.15 | 0.15 | 0.15 | 0.15 | 0.15 | 0.15 | 0.15 |
| ER$\alpha$-NTD(AF-1)/pcDNA3.1 | 0.00 | 0.01 | 0.02 | 0.05 | 0.10 | 0.20 | 0.40 | 0.80 | 1.00 | 1.20 | 1.40 | 1.60 | 1.80 | 2.00 | 0.10 | 0.00 |
| pcDNA3.1[5] | 2.25 | 2.24 | 2.23 | 2.20 | 2.15 | 2.05 | 1.85 | 1.45 | 1.25 | 1.05 | 0.85 | 0.65 | 0.45 | 0.25 | 2.20 | 2.30 |
| pGL4.74[6] | 0.05 | 0.05 | 0.05 | 0.05 | 0.05 | 0.05 | 0.05 | 0.05 | 0.05 | 0.05 | 0.05 | 0.05 | 0.05 | 0.05 | 0.05 | 0.05 |
| **Total DNA** [7] | **2.50** | **2.50** | **2.50** | **2.50** | **2.50** | **2.50** | **2.50** | **2.50** | **2.50** | **2.50** | **2.50** | **2.50** | **2.50** | **2.50** | **2.50** | **2.50** |

[1] For the gene transfection, a 6-well microplate was used for cell culture. $5 \times 10^5$ HeLa cells were delivered into each well (final volume: 2 mL).

[2] Under the condition XV and XVI with no full-length ER$\alpha$, it is necessary to assure that the expression plasmids of ER$\alpha$-NTD(AF-1)/pcDNA3.1 and pcDNA3.1 do not bind to the estrogen response element (ERE).

[3] Under the condition with no full-length ER$\alpha$, it is necessary to assess the basal activity derived from the endogenous ER$\alpha$.

[4] The abbreviation of 'full-length ER$\alpha$'.

[5] Empty plasmid vector which does not contain DNA of either full-length ER$\alpha$ or ER$\alpha$-NTD(AF-1).

[6] The vector encoding the luciferase reporter gene *hRluc* (*Renilla reniformis*).

[7] The limitation of total DNA amount, depending on the consumption of Lipofectamine$^{TM}$ 3000 transfection reagent.

ERα/pcDNA3.1 and FLAG-ERβ-NTD(AF-1)/pcDNA3.1. The assay was performed in duplicate at least three times under the conditions described above.

## Detailed inhibition tests of ERα-NTD(AF-1)

Gene transfections followed by luciferase reporter gene assays were carried out basically as described above. Test solutions were prepared using 6-well microplates with HeLa cells ($5 \times 10^5$ cells/well). First, for a detailed qualitative analysis of the inhibitory activity of non-tagged ERα-NTD(AF-1), an examination was carried out first by cellular transfection with serial amounts (0.01–2.00 μg/well) of the plasmid ERα-NTD(AF-1)/pcDNA3.1 as reported previously [20]. Further transfections were carried out using the following plasmids: full-length ERα/pcDNA3.1 (0.10 μg/well), 3 × ERE/pGL3 (0.30 μg/well), pcDNA3.1 (0.00–1.99 μg/well), and pGL4.74 (0.10 μg/well). After cell preparation, a luciferase reporter gene assay for 10 nM E2 was conducted on a 96-well microplate using the gene-transfected cells ($5 \times 10^4$ cells/well). The assessed activity of E2 was plotted for the original transfection amount (g) of ERα-NTD(AF-1) plasmid.

Another detailed quantitative analysis of the inhibitory action of ERα-NTD(AF-1) was performed, again following the previously reported method [20]. Using a 6-well microplate with HeLa cells ($5 \times 10^5$ cells/well), the expression plasmid of ERα-NTD(AF-1)/pcDNA3.1 was transfected at nine concentrations: 10, 31.6, 100, 316, 500, 1000, 1200, 1600, and 2000 ng/well, together with other expression plasmids of full-length ERα/pcDNA3.1 (0.05 μg/well), 3 × ERE/pGL3 (0.15 μg/well), pcDNA3.1 (0.25–2.25 μg/well), and pGL4.74 (0.05 μg/well). The luciferase reporter gene assay was conducted using a 96-well microplate containing gene-transfected cells ($5 \times 10^4$ cells/well), and the activity of E2 ($1.0 \times 10^{-13}$– $1.0 \times 10^{-6}$ M) was examined for full-length ERα.

From the dose–response curves of E2, different maximal activity levels were attained, depending on the transfection amounts of ERα-NTD(AF-1). Thus, each distinct maximal activity was evaluated precisely as a percentage. To further analyze the inhibitory activity, the maximal activities (%) were assessed based on the original transfection amount (g) of the ERα-NTD(AF-1) plasmid.

## Statistical analysis

All data are shown as means ± SD for the indicated number of separate experiments. The statistical significance of differences was determined by a two-sided Student's $t$-test. P-values less than 0.001 were considered significant.

## Immunocytochemical confocal microscopy analysis

To observe the locations of FLAG-tagged ERα-NTD(AF-1) and/or wild-type full-length ERα in the HeLa cells, they were expressed simultaneously or independently. FLAG-ERα-NTD(AF-1) was monitored with anti-FLAG antibody, while full-length intact ERα was monitored directly with ERα-specific antibody.

Cells were cultured in an Imaging Chamber CG 8 Well (Zell-Kontakt GmbH; Nörten-Hardenberg, Germany) at 37°C under 5% $CO_2$ for 48 h after transient transfection of the full-length ERα/pcDNA3.1 plasmid with or without the expression plasmid of FLAG-ERα-NTD(AF-1)/pcDNA3.1. For immunocytochemical labeling at 25°C, cells were first washed with phosphate-buffered saline (PBS) twice, then fixed with 4% paraformaldehyde and permeabilized using 0.2% Triton X-100 in PBS for 20 min. Cells were then washed in PBS three times for 10 min each time.

After blocking with 2% BSA and 0.1% Tween 20 in PBS overnight at 4°C, the cells were incubated with anti-FLAG antibody produced in a rabbit (Sigma-Aldrich, St. Louis, MO, USA) at a 1:800 dilution and anti-ERα (C-311) mouse monoclonal antibody IgG$_{2a}$ (Santa Cruz Biotechnology, Dallas, TX, USA) at a 1:1000 dilution in PBS-containing 2% BSA and 0.1% Tween 20 for 2 h. After washing with PBS three times for 10 min each time, the cells were further incubated for 60 min with Alexa Fluor 488-conjugated goat anti-rabbit IgG(H+L) secondary antibody (Thermo Fisher Scientific) (1:1000) for FLAG-ERα-NTD(AF-1), or Cy3-conjugated goat anti-mouse IgG(H+L) secondary antibody (Thermo Fisher Scientific) (1:1000) for full-length ERα. Finally, the cells were observed with an ECLIPSE TE2000-U confocal microscope (Nikon, Tokyo, Japan). A series of cellular treatments was conducted in an Imaging Chamber CG 8 Well to prepare cells for the cell culture, immunofluorescence staining, and imaging with a removable 8-well chamber.

## Results

### Activities of truncated ERα derivatives in the luciferase reporter gene assay

Dividing ERα into two parts—the N-terminal ERα-NTD(AF-1) and the C-terminal desNTD (AF-1)-ERα (Fig 1A)—we tested both in the luciferase reporter gene assay. Luciferase activity in the HeLa cells was measured approximately 48 h after their transfection. ERα-NTD(AF-1) was found to be completely devoid of transcription activation and exhibited absolutely no response to the ERα agonist 17β-estradiol (E2) (Fig 2). On the other hand, desNTD(AF-1)-ERα exhibited a definite transcription activation, although it was distinctly lower (approximately 34% lower) than the activity of full-length ERα (Fig 2B). The half-maximal effective concentration (EC$_{50}$) of E2 was very low, approximately 0.21 nM, showing a high activity for full-length ERα (Fig 2A). For desNTD(AF-1)-ERα, E2 also exhibited very high activity (EC$_{50}$ = 0.29 nM).

The 1:1 co-expression of N-terminal ERα-NTD(AF-1) and C-terminal desNTD(AF-1)-ERα resulted in an approximate 20% reduction in E2-induced transactivation compared to that mediated by desNTD(AF-1)-ERα alone (Fig 2B). This activity reduction was further enhanced by up to 44%, 62%, 73%, 85%, and 91% with the co-expression ratios of ERα-NTD(AF-1): desNTD(AF-1)-ERα = 2:1, 3:1, 4:1, 5:1, and 6:1, respectively. The result of dose-dependent inhibition clearly demonstrated that ERα-NTD(AF-1) acts as an efficient inhibitor of desNTD (AF-1)-ERα. The apparent IC$_{50}$ value was estimated to be 863 ng/well, where "well" indicates a well of the 6-well microplate originally used for the transfection of ERα-NTD(AF-1).

It is evident that the coexistence of ERα-NTD(AF-1) and desNTD(AF-1)-ERα does not lead to the amino acid sequential additivity necessary to reproduce ERα's full activity. On the contrary, the co-expression of ERα-NTD(AF-1) seemed to interfere with desNTD(AF-1)-ERα in some manner within the transcription mechanism. It should be noted that this activity interference or inhibition depends on the concentration of ERα-NTD(AF-1), and the target of this inhibition is desNTD(AF-1)-ERα, which consists of the ligand-binding domain LBD having AF-2.

### Effects of ERα-NTD(AF-1) on ERα transcription activity

The fact that ERα-NTD(AF-1) inhibited desNTD(AF-1)-ERα strongly suggested that untethered free ERα-NTD(AF-1) interacts with LBD(AF-2) but does not facilitate the construction of the binding site(s) for SRC coactivator(s). Since this led us to assume that ERα-NTD(AF-1) must inhibit also the full-length ERα, we decided to carry out an inhibition experiment to confirm this hypothesis. Inhibitory effectiveness was assessed by administering a wide range of concentrations (0.01–2.00 μg/well) of expression plasmid ERα-NTD(AF-1)/pcDNA3.1 or

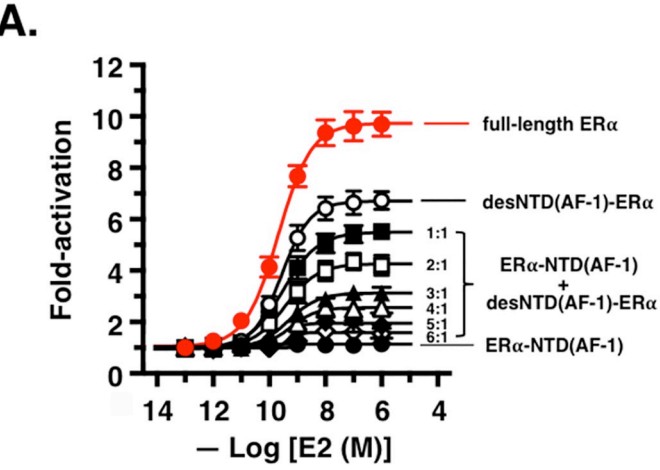

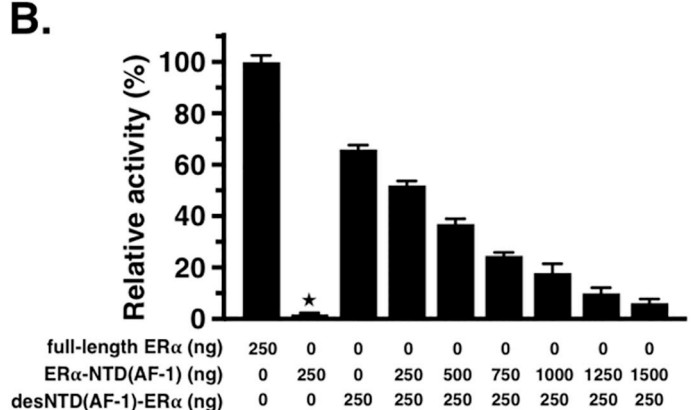

**Fig 2. Effects of co-expression of ER*α*-NTD(AF-1) and desNTD(AF-1)-ER*α* on E2 activity in the luciferase reporter gene assay. (A)** Dose–response curves for the activity of E2 ($1.0 \times 10^{-13}$– $1.0 \times 10^{-6}$ M) in HeLa cells transfected with expression plasmids of full-length ER*α* (●), desNTD(AF-1)-ER*α* (○), and ER*α*-NTD(AF-1) (•). desNTD(AF-1)-ER*α* activity was further evaluated in the presence of ER*α*-NTD(AF-1) under six different transfection conditions: with expression plasmid ratios of ER*α*-NTD(AF-1): desNTD(AF-1)-ER*α* = 1:1 (■), 2:1 (□), 3:1 (▲), 4:1 (△), 5:1 (♦), and 6:1 (◇). **(B)** Varied maximal activity (%) of E2 for desNTD(AF-1)-ER*α* shown by the bar graphs. Relative activities of full-length ER*α* and ER*α*-NTD(AF-1) were 100% and 0% (★), respectively, with a transfected plasmid amount of 250 ng/well in the 6-well microplate used for transfection.

FLAG-ER*α*-NTD(AF-1)/pcDNA3.1 for full-length ER*α*/pcDNA3.1 (0.05 *μ*g/well) and E2 (0.1, 1, or 50 nM) (Table 1).

This assessment revealed that the transcription activity of full-length ER*α* was reduced in a dose-dependent manner according to the transfection amounts of the ER*α*-NTD(AF-1) plasmid (Fig 3A). At the constant concentration of E2, increases in ER*α*-NTD(AF-1) gradually diminished the transcription activity of full-length ER*α*. Obviously, ER*α*-NTD(AF-1) worked to inhibit full-length ER*α*. This inhibitory activity of ER*α*-NTD(AF-1) was reproduced at all three E2 test concentrations: 0.1, 1, and 50 nM (Fig 3A). These results clearly revealed that the co-expressed synthetic ER*α*-NTD(AF-1) fragment competes with intact ER*α*-tethering NTD (AF-1), probably for the same site in ER*α*-LBD(AF-2). This competition for interaction

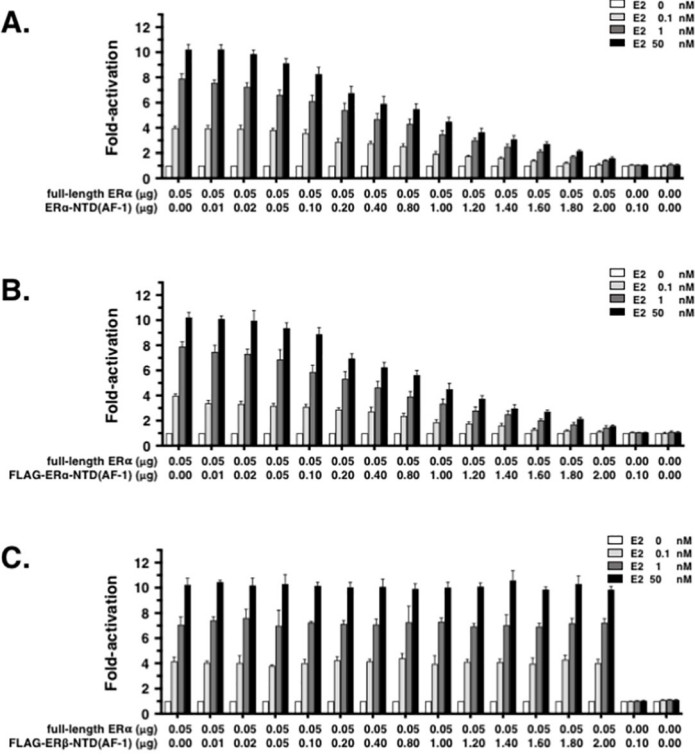

**Fig 3. Effects of untethered free ERα-NTD(AF-1) on the transcription activation activity of agonist E2 and receptor full-length ERα or ERβ.** Luciferase reporter assays carried out for ERα in HeLa cells using a constant concentration of E2 (0.1, 1, or 50 nM). The effects examined were ERα-NTD(AF-1) **(A)** and FLAG-ERα-NTD(AF-1) **(B)**. Plasmid amounts used for gene transfection (ng/well of a 6-well microplate; see Table 1) are noted under the bar graphs. **(C)** Luciferase reporter assays to validate the receptor specificity of ERα-NTD(AF-1) for ERα. The NTD(AF-1) domain of another estrogen receptor, ERβ, was tested for ERα in this series of assays, in which FLAG-ERβ-NTD(AF-1) was used instead of FLAG-ERα-NTD(AF-1). All other assay conditions were exactly the same as in **(B)**.

appears to reduce the luciferase transcription activity of full-length ERα in a manner dependent on the amount of transfected ERα-NTD(AF-1).

For an immunocytochemical study to confirm the interaction between ERα-NTD(AF-1) and full-length ERα, ERα-NTD(AF-1) was tagged with octapeptide DYKDDDDK, named FLAG-epitope peptide, at the N-terminus [14]. To evaluate the effect of this FLAG tag peptide on inhibitory activity, FLAG-ERα-NTD(AF-1) was also tested in the luciferase reporter assay. It immediately became clear that FLAG-ERα-NTD(AF-1) inhibits full-length ERα and E2 as well as non-tagged free ERα-NTD(AF-1) (Fig 3B). The presence or absence of the FLAG-tag was found to have no impact on the inhibitory activity of ERα-NTD(AF-1). It should be noted that untethered free ERα-NTD(AF-1) possesses an intrinsic ability to inhibit full-length ERα.

To evaluate the specificity of ERα-NTD(AF-1) in receptor inhibition, the NTD(AF-1) domain of another estrogen receptor, ERβ, was tested for full-length ERα. As shown in Fig 3C, FLAG-ERβ-NTD(AF-1) had no effect on the transcription activity of ERα. Thus, the inhibition of full-length ERα is particular and unique to ERα-NTD(AF-1) but not to ERβ-NTD(AF-1). The interaction between NTD(AF-1) and LBD(AF-2) in ERα appears to be highly specific.

## Detailed analysis of ERα inhibition by ERα-NTD(AF-1)

The inhibitory activity of ERα-NTD(AF-1) was examined using both a qualitative analytical method and a quantitative analytical method. First, the inhibitory activity of ERα-NTD(AF-1)

was verified qualitatively in a luciferase reporter gene assay. Prior to the assay, HeLa cells were transfected with serial amounts of the plasmid of non-tagged free ERα-NTD(AF-1) ($1.0 \times 10^{-8}$ to $2.0 \times 10^{-6}$ g/well of a 6-well microplate). The activity of 10 nM E2 was evaluated for full-length ERα using cells treated with these various amounts of ERα-NTD(AF-1). At 10 nM, E2 exhibited 90–95% transcriptional activity for ERα. When this amount of E2 was tested in the cells treated with ERα-NTD(AF-1), the activity decreased according to the amount of ERα-NTD(AF-1) (Fig 4A). It is evident that untethered free ERα-NTD(AF-1) functions to inhibit full-length ERα. From the dose–response curve, the apparent $IC_{50}$ value was estimated to be 836 ng/well, equivalent to the activity obtained for ERα-NTD(AF-1) in the inhibition of desNTD(AF-1)-ERα (see above).

We next evaluated the transcriptional activity of E2 quantitatively in the presence of non-tagged free ERα-NTD(AF-1) across nine transfection amounts: 10, 31.6, 100, 316, 500, 1000, 1200, 1600, and 2000 ng/well (where "well" indicates a well of the 6-well microplate used for transfection). As shown in Fig 4B, each dose–response curve of E2 ($1.0 \times 10^{-13}$– $1.0 \times 10^{-6}$ M) reached a different level of maximal activation activity. The activity levels varied depending on the amounts of ERα-NTD(AF-1) plasmid used for transfection, as observed for the inhibition of desNTD(AF-1)-ERα. When we show the activity as a percentage of the full activity (100%) of neat full-length ERα, the maximal activation activity level attenuated gradually from 96% to 14% (Fig 4B, Table 2). In other words, when the transfection amount increased, ERα-NTD (AF-1) sharply reduced the maximal activity of E2 (Fig 4B).

The reduction of maximal activation activity signifies a decrease in the active receptor population. In the luciferase reporter gene assays carried out for this test, E2 showed almost the same $EC_{50}$ values (Fig 4B, Table 2). The average $EC_{50}$ value was 0.296 nM, and the intrinsic $EC_{50}$ value of E2 was 0.277 nM with no ERα-NTD(AF-1) transfection (Table 2). Together, these results implied that ERα-NTD(AF-1) reduces the activity of full-length ERα without competing with E2 itself at the binding site. Apparently, ERα-NTD(AF-1) diminished the proper functioning of ERα receptor molecules. It is evident that ERα-NTD(AF-1) is a kind of non-competitive inhibitor of full-length ERα. As the transfection amount of ERα-NTD(AF-1) increased, the maximal activity of ERα/E2 decreased.

These results made it impossible to conduct a Schild plot analysis, which is applicable only for competitive inhibitors [21, 22]. Since it was obvious that there is a dose–response relationship between the transfection amount of inhibitor ERα-NTD(AF-1) and the maximal activity of ERα/E2 (Table 2), we generated a dose–response curve to assess the $IC_{50}$ of ERα-NTD(AF-1) (Fig 4C). This value was calculated to be 631 ng/well, where, as noted above, "well" refers to a well of the 6-well microplate originally used for the transfection of the ERα-NTD(AF-1) plasmid. This $IC_{50}$ value was roughly equivalent to that obtained from the qualitative luciferase reporter gene assay (Fig 4A). These results demonstrated that separately synthesized ERα-NTD(AF-1) effectively inhibited the transcription activation activity of E2 for full-length ERα, but in a non-competitive manner. In conclusion, untethered free ERα-NTD(AF-1) is a strong inhibitor of full-length ERα.

## Nuclear translocation of ERα-NTD(AF-1) by binding to ERα

Separately expressed ERα-NTD(AF-1) exhibited strong inhibitory activity in the transcription activation mechanism of full-length ERα. This outcome implied that untethered free ERα-NTD(AF-1) binds to full-length ERα, probably to its C-terminal LBD. The resulting complex between the ERα-NTD(AF-1) fragment and full-length ERα abrogates ERα transcription, potentially halting the activity at a crucial early stage of gene expression. Subsequently, we intended to demonstrate these processes immunocytochemically by staining the ERα-NTD

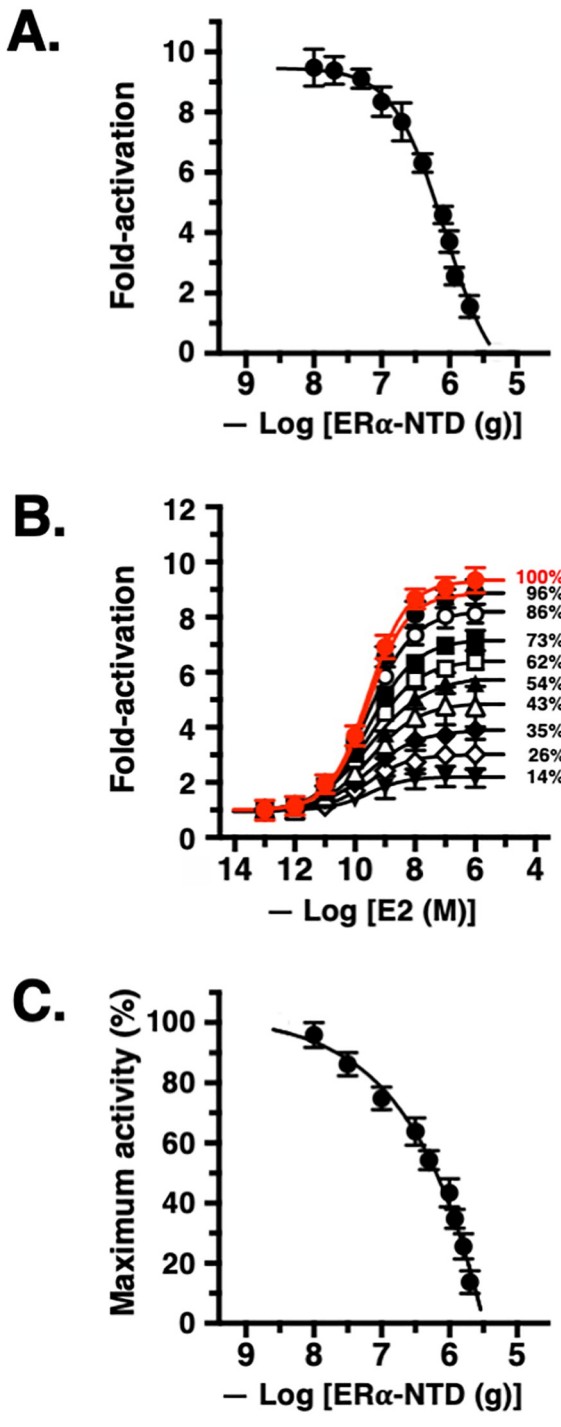

**Fig 4. Inhibition tests in the luciferase reporter gene assay for E2 and ER*α* in the presence of ER*α*-NTD(AF-1) fragment peptide (1–180). (A)** Dose–response curves illustrating the residual activity of 10 nM E2 in ER*α* in the presence of HeLa cells transfected with serial amounts of the expression plasmid encoding ER*α*-NTD(AF-1). **(B)** Varied maximal activity of E2 ($1.0 \times 10^{-13}$–$1.0 \times 10^{-6}$ M) in the absence (●) and presence of nine different transfection amounts of the ER*α*-NTD(AF-1) plasmid [10 (●)), 31.6 (○), 100 (■), 316 (□), 500 (▲), 1000 (△), 1200 (♦), 1600 (◇), and 2000 (▼) ng/well of the 6-well microplate used for transfection], and **(C)** dose–response curve depicting the relationship between the amount of the ER*α*-NTD(AF-1) plasmid used for cellular transfection and the maximal activity of E2 in the luciferase reporter gene assay.

**Table 2. Analyses of inhibitory activity of ER$\alpha$-NTD(AF-1) for ER$\alpha$/17$\beta$-estradiol (E2) in the luciferase reporter gene assay.**

| Inhibitor concentration [ER$\alpha$-NTD(AF-1)] (ng/well)[1] | Agonist 17$\beta$-estradiol (E2) | |
|---|---|---|
| | Maximal activity (%) | EC$_{50}$ (nM) |
| 0 | 100 | 0.277 |
| 10 | 96 | 0.290 |
| 31.6 | 86 | 0.289 |
| 100 | 76 | 0.297 |
| 316 | 62 | 0.309 |
| 500 | 54 | 0.298 |
| 1,000 | 43 | 0.318 |
| 1,200 | 35 | 0.308 |
| 1,600 | 26 | 0.280 |
| 2,000 | 14 | 0.273 |
| **Average EC$_{50}$** [2] | 0.296 | |

[1] The reaction solutions were prepared in the well of a 6-well microplate.

[2] The EC$_{50}$ values of E2, which were estimated in the presence of inhibitor ER$\alpha$-NTD(AF-1), were averaged.

(AF-1) fragment and full-length ER$\alpha$ separately. ER$\alpha$-NTD(AF-1) was thus tagged with FLAG-epitope peptide at the N-terminus for detection with the polyclonal anti-FLAG antibody. As demonstrated above, the presence of the FLAG-tag had no influence on the inhibitory activity of ER$\alpha$-NTD(AF-1).

ER$\alpha$ was stained by a mouse monoclonal antibody (IgG$_{2a}$) raised against ER$\alpha$. This anti-ER$\alpha$ monoclonal antibody specifically recognizes the C-terminal region (495–595) of ER$\alpha$. Therefore, the separately synthesized N-terminal fragment ER$\alpha$-NTD(AF-1) does not respond to this antibody. In fact, it was confirmed in advance that the anti-FLAG antibody and anti-human ER$\alpha$ antibody do not cross-react at all between ER$\alpha$-NTD(AF-1) and full-length ER$\alpha$.

First, when neither FLAG-ER$\alpha$-NTD(AF-1) nor ER$\alpha$ was transfected into the HeLa cells, it was confirmed that the primary antibodies used—namely, anti-FLAG polyclonal antibody and anti-ER$\alpha$ monoclonal antibody—did not show any background staining (Fig 5A). FLAG-ER$\alpha$-NTD(AF-1) peptide was detected only in the cytoplasm (Fig 5B) when FLAG-ER$\alpha$-NTD(AF-1) was solely transfected and expressed in the largest transfection amount (0.5 $\mu$g/well, where "well" refers to a well of the Imaging Chamber CG 8 Well used for the cell culture for image analysis). This introductory amount of FLAG-ER$\alpha$-NTD(AF-1) is equivalent to the amount necessary for the full inhibition of ER$\alpha$. There was absolutely no staining at the cell nuclei, only exclusive staining at the cytoplasm (Fig 5B). Such cytoplasmic localization, i.e., occurrence only in the cytoplasm, of FLAG-ER$\alpha$-NTD(AF-1) was retained throughout the experiment (more than 3 days). By contrast, when full-length ER$\alpha$ was solely transfected, its immunostaining was observed exclusively in the cell nucleus from the very start of process (Fig 5C).

Full-length ER$\alpha$ includes the nuclear localization signal (NLS) at the region named *hinge*, which lies between the DBD and the LBD (Fig 1A) [23, 24]. It should be noted, however, that ER$\alpha$-NTD(AF-1) does not have such an NLS or any NLS-like signals. If FLAG-ER$\alpha$-NTD(AF-1) was expressed simultaneously with full-length ER$\alpha$ in the same cell preparation, full-length ER$\alpha$ would transfer this synthetic FLAG-ER$\alpha$-NTD(AF-1) from the cytoplasm into the cell nucleus, forming an ER$\alpha$-NTD(AF-1)•ER$\alpha$ complex. Since co-expressed synthetic ER$\alpha$-NTD (AF-1) is tagged with the N-terminal FLAG-epitope peptide, their nuclear translocation would

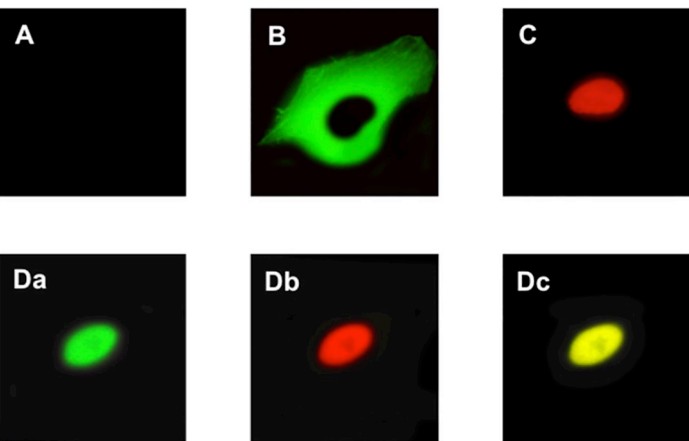

**Fig 5. Immunocytochemical localization of FLAG-ERα-NTD(AF-1) and full-length ERα in HeLa cells. (A)**
Control staining with rabbit polyclonal anti-FLAG primary antibody, followed by Alexa Fluor 488-conjugated goat
anti-rabbit IgG(H+L) secondary antibody for FLAG-ERα-NTD(AF-1). Similar control staining was performed using
anti-ERα mouse monoclonal primary antibody and Cy3-conjugated goat anti-mouse IgG(H+L) secondary antibody,
which were used for full-length ERα (data not shown). **(B)** Only FLAG-ERα-NTD(AF-1) was expressed in the cells.
The peptide was detected as a green color. **(C)** Only full-length ERα was expressed in the cells. ERα was detected as a
red color. **(D)** Evaluation of co-localization carried out by co-staining for FLAG-ERα-NTD(AF-1) and full-length ERα.
After exclusive detection of FLAG-ERα-NTD(AF-1) (green, panel **Da**) and full-length ERα (red, panel **Db**), merged
immunolabeling was observed (yellow, panel **Dc**).

be monitored immunocytochemically using each specific antibody. Conversely, this transloca-
tion becomes a demonstration of the intermolecular interaction between FLAG-ERα-NTD
(AF-1) and full-length ERα. Such actual experimental proof was indeed realized for the unteth-
ered free ERα-NTD(AF-1) fragment in this study.

Fig 5Da shows the anti-FLAG immunostaining of FLAG-ERα-NTD(AF-1) in the HeLa
cells when co-expressed together with full-length ERα. Alexa Fluor 488's green immunolabel-
ing for FLAG-ERα-NTD(AF-1) was observed only in the cell nucleus and not in the cytoplasm.
In spite of this green immunostaining, the same nucleus was stained red by the observation at
550 nm (Fig 5Db). This is because of the Cy3 immunolabeling of full-length ERα. As a result,
as shown in Fig 5Dc, merging of the green and red images yielded an entirely yellow immuno-
labeling image. These observations confirmed that the full-length ERα and FLAG-ERα-NTD
(AF-1) peptide were completely co-localized in the cell nucleus, clearly revealing that sepa-
rately expressed untethered free FLAG-ERα-NTD(AF-1) interacts and combines with full-
length ERα to move from the cytoplasm into the nucleus.

## Discussion

### Conformation of ERα-NTD(AF-1) for interaction with LBD(AF-2)

In the present study, two distinct experimental results indicated that the N-terminal domain of
ERα, namely, ERα-NTD(AF-1), interacts directly with the C-terminal ligand-binding domain
ERα-LBD(AF-2). These results are: (i) separately expressed untethered free ERα-NTD(AF-1)
inhibited full-length ERα without competing with the agonist E2; and (ii) separately expressed
untethered free ERα-NTD(AF-1) was translocated from the cytoplasm into the cell nucleus
when co-expressed with full-length ERα. These findings were obtained based on the discovery
by Yi et al. [5] that ERα-NTD(AF-1) is adjacent to LBD(AF-2) in the ERα•DNA complex and
on our subsequent speculation that NTD(AF-1) and LBD(AF-2) must interact directly with

each other. Since ERβ-NTD(AF-1) did not inhibit full-length ERα, at least in terms of estrogen receptors, this NTD(AF-1) and LBD(AF-2) are specific to the ERα receptor.

The question of which part of NTD(AF-1) is adjacent to LBD(AF-2) is not answered in the literature [5]. Our structural analysis suggested the loop-containing region(s) of H3–H4 and/ or H8–H9 of ERα-LBD(AF-2) as a site or sites to which NTD(AF-1) binds (Fig 1B). In general, the structure of NTD(AF-1) in various members of the nuclear receptor superfamily is characterized by a conformation lacking a stable three-dimensional (3D) structure and having very high flexibility [25–27]. The structure of ERα-NTD(AF-1) was also reported to be flexible and disordered [28, 29].

It has recently become feasible to predict the 3D structures of proteins with considerably high accuracy by using a deep-learning artificial intelligence (AI) system. In particular, the AI system AlphaFold can depict the 3D structure of a protein from its amino acid sequence [30, 31]. According to the AlphaFold website, "The AlphaFold data base provides open access to over 200 million protein structure predictions to accelerate scientific research" [32]. When we used this database to determine and analyze the 3D structure of full-length ERα, AlphaFold presented one structure (P03372) from 429 structures in the protein data bank (PDB) [33]. We further analyzed the predicted structure in detail using the software package DiscoveryStudio. As reported in PDB, the DNA-binding domain (DBD: 181–263) and ligand-binding domain (LBD: 303–552) possess the best-known ordered 3D structures (Fig 6A). Most of the other domains [NTD(AF-1), *hinge*, and F-domain] have intrinsically disordered structures, i.e., they lack α-helices and β-strands.

The only exception is the α-helix in NTD(AF-1). At positions 11–18, the secondary structure α-helix was found to lie in the two helical turns (Fig 6A and 6B). In addition, this short α-helix was found to interact with ERα-LBD(AF-2) (Fig 6B). There are two hydrogen bonds, namely, Gly11-αNH···H12 Glu542-δCO (1.79 Å) and Ile18-αCO···H3 Lys362-εNH (2.59 Å), and three hydrophobic bonds, namely, Leu14···H3 Ile358 (5.11 Å), Leu14···H12 Leu539 (3.27 Å), and Leu15···H3 Ile358 (5.32 Å). These bonds appear to confirm a solid interaction between the short α-helix in NTD(AF-1) and the C-terminal LBD(AF-2). As a consequence, the short α-helix appears to interact with several amino acid residues in H3 and H12 present in ERα-LBD(AF-2).

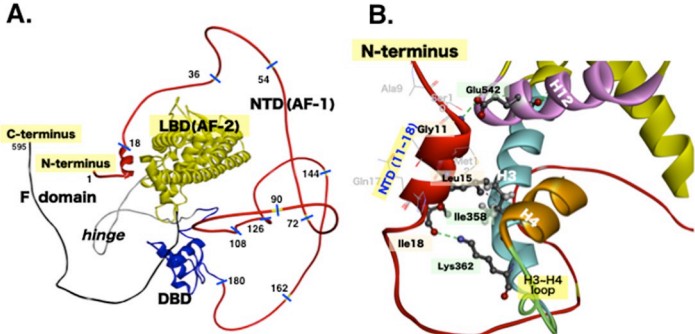

**Fig 6. AlphaFold structural prediction of human estrogen receptor ERα. (A)** Predicted 3D structure of the full-length ERα protein (1–595). This AlphaFold structure prediction is available on the AlphaFold website: https://alphafold.ebi.ac.uk/entry/P03372. P03372, a predicted 3D structure of full-length ERα, was further analyzed in detail using the software package DiscoveryStudio. The designated domain structures are the NTD(AF-1), DBD, *hinge*, LBD, and F domain, arranged from the N-terminus to the C-terminus. **(B)** Structural details of the domain–domain interaction between the N-terminal α-helix (11–18) in ERα-NTD(AF-1) and the H3-loop-H4 region adjacent to H12/ AF-2 in ERα-LBD.

It should be noted that the H3-loop-H4 region on the LBD surface area is adjacent to the H12/AF-2 site in ER*α* (Fig 1B). It is highly likely that the separately expressed untethered free ER*α*-NTD(AF-1) peptide is in a conformation similar to that of NTD(AF-1) in full-length ER*α*. In addition to the N-terminal *α*-helix (11–18) of intact tethered ER*α*-NTD(AF-1), the untethered free ER*α*-NTD(AF-1) peptide would also possess an *α*-helix (11–18) to interact with the H3-loop-H4 and LBD H12/AF-2 region. This implies that untethered free ER*α*-NTD (AF-1) peptide and the intact tethered ER*α*-NTD(AF-1) compete to interact with these regions. If the interaction between the untethered free ER*α*-NTD(AF-1) peptide and the H3-loop-H4 and LBD H12/AF-2 region fails to establish a proper conformation of the binding site for the coactivator proteins, such interaction competition must result in the inhibition of the transcription activation activity of intact full-length ER*α*. It is highly likely that, when NTD (AF-1) is properly linked to the central DBD domain, the *α*-helix structural unit at positions 11–18 plays a crucial role in ER*α*'s recruitment of the SRC coactivator proteins.

## Nuclear translocation of ER*α*-NTD(AF-1) by interacting with LBD(AF-2)

In Fig 7, a putative molecular mechanism of nuclear translocation of untethered free ER*α*-NTD(AF-1) is depicted together with its mechanism of inhibitory activity against full-length ER*α*. The cytoplasmic localization is characteristic of the 1–180 peptide fragment of ER*α*, specifically untethered free ER*α*-NTD(AF-1), when expressed alone (case I) (Fig 7). This was demonstrated by an immunocytochemical staining study using FLAG-tagged ER*α*-NTD(AF-1) (Fig 5). ER*α*-NTD(AF-1) does not have an NLS signal, unlike full-length ER*α*, which can be translocated from the cytoplasm into the nucleus due to the signal (case III) (Fig 7). Straddling the C-terminal LBD(AF-2) of full-length ER*α*, untethered free ER*α*-NTD(AF-1) or its FLAG-tagged ER*α*-NTD(AF-1) peptide can move into the nucleus despite the absence of the NLS signal (case II) (Fig 7).

Untethered free ER*α*-NTD(AF-1) peptide forms an ER*α*-NTD(AF-1)•ER*α* complex by binding to the region involving H3-loop-H4 and H12/AF-2 in ER*α*-LBD (Fig 6B). It is unclear whether the interaction mode of untethered free ER*α*-NTD(AF-1) peptide with ER*α*-LBD(AF-2) is exactly the same as that of intact tethered ER*α*-NTD(AF-1) with its own ER*α*-LBD(AF-2).

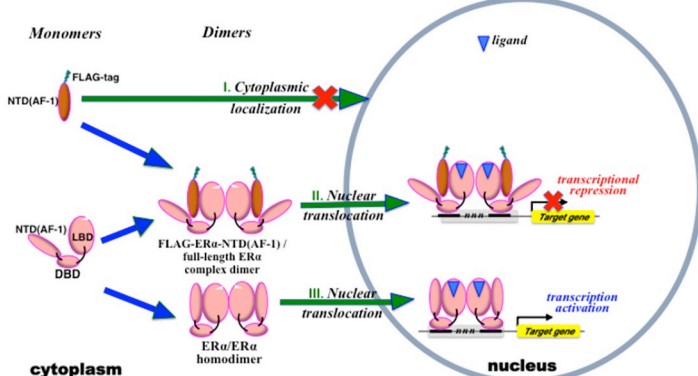

**Fig 7. Outline flowchart of the nuclear translocation of ER*α*-NTD(AF-1)/ER*α* complex and transcription inhibition resulting from complexation in the cell.** The figure shows only the mechanisms for FLAG-tagged ER*α*-NTD(AF-1), but exactly the same mechanisms are valid for non-tagged ER*α*-NTD(AF-1). Expressed FLAG-ER*α*-NTD (AF-1) has no nuclear localization signal (NLS), and therefore it remains in the cytoplasm (case I). When co-expressed together with full-length ER*α*, the resulting FLAG-ER*α*-NTD(AF-1)•ER*α* complex migrates from the cytoplasm into the nucleus (case II). This complex is not active in the luciferase reporter gene assay. ER*α* dimerized in the cytoplasm moves into the nucleus (case III).

However, this interaction certainly resulted in the inhibition of the transcription activity of full-length ER$\alpha$.

The complexation between the untethered free ER$\alpha$-NTD(AF-1) peptide and the full-length ER$\alpha$ protein would enable the translocation of untethered free ER$\alpha$-NTD(AF-1) into the nucleus: that is, full-length ER$\alpha$ transports ER$\alpha$-NTD(AF-1) from the cytoplasm into the nucleus (case II) (Fig 7). Exactly the same nuclear translocation of ER$\alpha$-NTD(AF-1) was brought about by the truncated ER$\alpha$-derivative desNTD(AF-1)-ER$\alpha$ (data not shown).

Full-length ER$\alpha$ exists as a dimer via the specific dimerization that first occurs at the H10 region of LBD(AF-2) (Fig 1B). As a result, there are two types of ER$\alpha$-NTD(AF-1)•ER$\alpha$ complex: one with complexation at both of the LBD(AF-2) domains as shown in case II (Fig 7) and one with complexation at only one of the two LBD(AF-2) domains in a dimer (not shown). In both cases, ER$\alpha$-NTD(AF-1) is translocated into the nucleus.

### Inhibitory mode of untethered free ER$\alpha$-NTD(AF-1)

It should be noted that the ER$\alpha$-NTD(AF-1)•ER$\alpha$ complexes possess a sound, undamaged DNA-binding domain DBD in the full-length ER$\alpha$ protein, and therefore complexes carrying untethered free ER$\alpha$-NTD(AF-1) can bind to DNA through conserved estrogen response element sequence motifs. However, as demonstrated in this study, the formed DNA complexes fail to properly recruit the coactivator protein SRC, likely because of the presence of "additive" untethered free ER$\alpha$-NTD(AF-1) peptide. A physiologically inactive ER$\alpha$-NTD(AF-1)•ER$\alpha$ complex has two NTD(AF-1) moieties: a co-expressed untethered free NTD(AF-1) peptide and an intact ER$\alpha$-tethered NTD(AF-1) moiety. The challenge lies in elucidating why intact NTD(AF-1) functions as a stimulator while untethered free NTD(AF-1) functions as an inhibitor.

Even if untethered free ER$\alpha$-NTD(AF-1) peptide is able to construct a binding site for the SRC coactivator, the intact ER$\alpha$-NTD(AF-1) domain, a larger protein unit having 180 amino acid residues, might obstruct or impede the binding of SRC protein due to its steric hindrance. Perhaps the intact ER$\alpha$-NTD(AF-1) plays some roles in preventing the acceptance of coactivator proteins. On the other hand, since only a small part of the co-expressed untethered free NTD(AF-1) peptide, namely, the $\alpha$-helix (11–18), appears to bind to ER$\alpha$-LBD (Fig 6B), the large remaining disordered part may also play a role in such hindrance or interference.

### Inquiry into the intrinsic functional role of ER$\alpha$-NTD(AF-1)

DesNTD(AF-1)-ER$\alpha$ is almost equivalent to the native ER$\alpha$ isoform named ER$\alpha$46, which was reported to be a weak transcriptional activator compared to ER$\alpha$ [34, 35]. ER$\alpha$46 is 4 amino acids longer than desNTD(AF-1)-ER$\alpha$ at its N-terminus. As an E2-inducible activator, ER$\alpha$46 has been reported to be 20–100% as efficient as full-length ER$\alpha$ in assays using various cells and promoter reporter genes. Recently, ER$\alpha$46 was reported to be involved in the growth and metastasis of breast cancer cells [36]. Thus, what is its physiological role in breast cancer cells?

In this study, desNTD(AF-1)-ER$\alpha$ exhibited ca. 65% activity of full-length ER$\alpha$. This result implies that ER$\alpha$ does not necessarily require NTD(AF-1) for its transcription activity. ER$\alpha$46 shares an identical DNA sequence with wild-type full-length ER$\alpha$ from positions 174 to 595 [37, 38], and thus they share only AF-2 of the two activation function motifs. In the transcription activation mechanism, however, it is necessary for both ER$\alpha$46 and full-length ER$\alpha$ to bind to the co-activator SRC and additionally to the secondary co-activator, p300. To recruit these co-activator proteins, full-length ER$\alpha$ utilizes both AF-1 and AF-2. ER$\alpha$46 possesses only AF-2, and thus it has to utilize its LBD(AF-2) domain exclusively to recruit an SRC co-activator protein.

On the recruitment of the p160 coactivator proteins, SRCs utilize a region known as the "nuclear receptor interaction domain" (NID). NID contains three helical motifs sharing an LXXLL core consensus sequence, referred to as the nuclear receptor NR boxes I, II, and III. Research suggests that distinct NR boxes interact differently with various nuclear receptors, indicating that a single NR box is sufficient for activation by the estrogen receptor ERα [39]. This was verified for the SRC-1 and SRC-2 NR boxes; ERα showed a distinct preference for NR box II [40]. In contrast, SRC-3 utilizes both NR boxes I and II simultaneously for ERα, although ERα exhibits a bigger preference (approximately 2-fold) for NR box I than for box II [40], In this scenario, it is likely that NTD(AF-1) and LBD(AF-2) cooperate to form a specific binding site for the recruitment of these SRC proteins.

## Perspectives on the structural role of N-terminal NTD(AF-1) domain

Because of the lack of NTD(AF-1), ERα46 should recruit SRC proteins differently from full-length ERα. In this case, ERα46 uses its LBD(AF-2) for recruitment. It should be noted that, when ERα46 is co-expressed with full-length ERα, ERα46 inhibits ERα exclusively by forming their heterodimer [41, 42]. In the ERα46•ERα heterodimer, ERα46 lacks the adjacent NTD (AF-1) domain in its C-terminal LBD(AF-2), whereas the partner ERα has NTD(AF-1) and LBD(AF-2) naturally adjacent to each other. Consequently, the heterodimer ERα46•ERα possesses non-equivalent SRC-binding sites, recruiting the SRC coactivator for each. However, these coactivators probably exhibit spatial asymmetry, leading to a failure to recruit the secondary coactivator p300 or to form a transcriptionally productive p160•p300 complex.

Another ERα isoform, named ERα-LBD, has recently been identified in MCF-7 cells and reportedly promotes breast cancer and endocrine resistance [43]. This ERα-LBD isomer would form also a heterodimer with full-length ERα, and the resulting ERα-LBD•ERα heterodimer would be nonproductive.

Formation of ERα heterodimers such as ERα46•ERα and ERα-LBD•ERα might hinder the turnover of intact full-length ERα, impacting or promoting the proliferation of cancer cells. Thus, ERα-NTD(AF-1) appears to be a molecular determinant of the cellular state: its presence in the full-length natural estrogen receptor ERα determines its physiological role as a nuclear receptor, while its absence in the isomer ERα46 or ERα-LBD might determine its pathophysiological role as a molecular carcinogenic or tumor progressive factor.

The structural analysis of the ERα-coactivator complex on DNA suggests the N/C-intramolecular interaction between the N-terminal ERα-NTD(AF-1) and the C-terminal ERα-LBD (AF-2). We then demonstrated in the present study the intermolecular interaction between the untethered free ERα-NTD(AF-1) and the LBD(AF-2) domain of full-length ERα. Since this is the indirect evidence, however, it is now crucial to prove the interaction directly using biochemical methods. We are preparing to conduct an experiment to provide such evidence.

## Conclusion

Based on structural information about the DNA-bound ERα-coactivator complex, we revealed that, to construct the binding sites of coactivator SRC proteins, ERα-NTD(AF-1) interacts with the region including the H3-loop-H4 and H12/AF-2 in the C-terminal ERα-LBD(AF-2) domain. This N/C-intramolecular interaction was demonstrated by two pieces of evidence: untethered free ERα-NTD(AF-1) inhibited the transcription activity of full-length ERα in the luciferase reporter gene assay, and the cytoplasm-to-nucleus translocation of ERα-NTD(AF-1) was immunocytochemically detected only when co-expressed with full-length ERα. The present results offer insights into the molecular design of ERα-specific inhibitors, independent of receptor antagonist.

## Author Contributions

**Conceptualization:** Yasuyuki Shimohigashi.

**Data curation:** Xiaohui Liu, Yutaka Matsuyama, Makiko Sugiyama, Keitaro Suyama, Yasuyuki Shimohigashi.

**Formal analysis:** Xiaohui Liu, Yutaka Matsuyama, Makiko Sugiyama, Yasuyuki Shimohigashi.

**Funding acquisition:** Xiaohui Liu, Yasuyuki Shimohigashi.

**Methodology:** Xiaohui Liu, Keitaro Suyama, Takeru Nose, Miki Shimohigashi, Yasuyuki Shimohigashi.

**Project administration:** Yasuyuki Shimohigashi.

**Resources:** Xiaohui Liu, Yasuyuki Shimohigashi.

**Software:** Xiaohui Liu, Keitaro Suyama, Yasuyuki Shimohigashi.

**Supervision:** Xiaohui Liu, Yasuyuki Shimohigashi.

**Validation:** Xiaohui Liu, Keitaro Suyama, Takeru Nose, Miki Shimohigashi, Yasuyuki Shimohigashi.

**Visualization:** Xiaohui Liu.

**Writing – original draft:** Xiaohui Liu, Yasuyuki Shimohigashi.

**Writing – review & editing:** Xiaohui Liu, Miki Shimohigashi, Yasuyuki Shimohigashi.

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
