## [Decision Letter · Decision Letter 0]

30 Jul 2024

PONE-D-24-22407The N-terminal activation function AF-1 domain of ERα interacts directly with the C-terminal AF-2-holding ligand-binding domain to recruit the coactivator proteinsPLOS ONE

Dear Dr. Liu,

Thank you for submitting your manuscript to PLOS ONE. After careful consideration, we feel that it has merit but does not fully meet PLOS ONE’s publication criteria as it currently stands. Therefore, we invite you to submit a revised version of the manuscript that addresses the points raised during the review process.

Please submit your revised manuscript by Sep 13 2024 11:59PM. If you will need more time than this to complete your revisions, please reply to this message or contact the journal office at plosone@plos.org. Please include the following items when submitting your revised manuscript:A rebuttal letter that responds to each point raised by the academic editor and reviewer(s). You should upload this letter as a separate file labeled 'Response to Reviewers'.A marked-up copy of your manuscript that highlights changes made to the original version. You should upload this as a separate file labeled 'Revised Manuscript with Track Changes'.An unmarked version of your revised paper without tracked changes. You should upload this as a separate file labeled 'Manuscript'.If applicable, we recommend that you deposit your laboratory protocols in protocols.io to enhance the reproducibility of your results. Protocols.io assigns your protocol its own identifier (DOI) so that it can be cited independently in the future. For instructions see: https://journals.plos.org/plosone/s/submission-guidelines#loc-laboratory-protocols. Additionally, PLOS ONE offers an option for publishing peer-reviewed Lab Protocol articles, which describe protocols hosted on protocols.io. Read more information on sharing protocols at https://plos.org/protocols?utm_medium=editorial-email&utm_source=authorletters&utm_campaign=protocols.

We look forward to receiving your revised manuscript.

Kind regards,

Jin-Ming Zhou

Academic Editor

PLOS ONE

3 Thank you for stating the following financial disclosure: 

 [This work was supported by the Japan Society for the Promotion of Science (JSPS) KAKENHI; grant nos. 25740024, 15K00557, 19K12340 and 22K12395 (to X.L.) and 22221005 and 15H01741 (to Y.S.). This work was also supported in part by a Health and Labour Sciences Research Grant for Research on the Risk of Chemical Substances from the Ministry of Health, Labor, and Welfare of Japan, under the contract/grant no. H20-Chemistry-General-003 (to Y.S.).].  

5. We note that Figures 1, 6 and 7 in your submission contain [map/satellite] images which may be copyrighted. All PLOS content is published under the Creative Commons Attribution License (CC BY 4.0), which means that the manuscript, images, and Supporting Information files will be freely available online, and any third party is permitted to access, download, copy, distribute, and use these materials in any way, even commercially, with proper attribution. For these reasons, we cannot publish previously copyrighted maps or satellite images created using proprietary data, such as Google software (Google Maps, Street View, and Earth). For more information, see our copyright guidelines: http://journals.plos.org/plosone/s/licenses-and-copyright.

1. You may seek permission from the original copyright holder of Figures 1, 6 and 7 to publish the content specifically under the CC BY 4.0 license.  

Additional Editor Comments (if provided):

Reviewers' comments:

Reviewer's Responses to Questions

**Comments to the Author**

1. Is the manuscript technically sound, and do the data support the conclusions?

Reviewer #1: Yes

2. Has the statistical analysis been performed appropriately and rigorously? 

Reviewer #1: I Don't Know

3. Have the authors made all data underlying the findings in their manuscript fully available?

Reviewer #1: Yes

4. Is the manuscript presented in an intelligible fashion and written in standard English?

Reviewer #1: Yes

5. Review Comments to the Author

Reviewer #1: REVIEW WITH MINOR REVISION COMMENT ADDED.

Liu et al have used a previously published cryoEM structure of human ERalpha, which shows that the N-terminal domain (NTD) and ligand binding domain (LBD) at the C-terminus are spatially close, to investigate the role in transcriptional activation of ERalpha of the interaction of N-terminal domain (NTD) with the ligand binding (LBD).

Their key finding is that the NTD from ERalpha can inhibit transcriptional activation of ERalpha. Interestingly, the NTD from ERbeta does not inhibit ERalpha. Thus, the inhibition of activation of ERalpha is specific to the NTD of ERalpha.

Then Liu et al provide a structural analysis of the interaction of the NTD from ERa with the full-length ERa using the cryoEM structure of ERalpha. This leads Liu et al to propose that binding of the ERalpha NTD to the LBD of ERalpha interferes with binding of SRC co-activator proteins, which in turn inhibits transcriptional activation of ERalpha. This is a reasonable hypothesis.

This is a very thorough and detailed study, which will be of much interest to endocrinologists studying ERalpha. The experiments are well-done. The data and figures are solid.

Minor Revision comments.

However, the writing needs editing from two perspectives.

First, in the Abstract: What is ERalpha-NTD(AF-1)? Is it the N-terminal domain [residues 1-180], without a DNA or ligand binding domain, or is it the full ERalpha [residues 1-595]? What is desNTD(AF-1)-ERa? Is it a truncated ERalpha [residues 181-595], containing the DBD, hinge and LBD?

I suggest adding explicit definitions with the residues to desNTD(AF-1)-ERa and ER-NTD(AF-1) to the Abbreviations Section.

Also The text in line 279-285 on ERalpha46 could be condensed and moved to the end Discussion section. Keep the focus on the relevance for full-length ERalpha, which is the important biological estrogen receptor.

6. PLOS authors have the option to publish the peer review history of their article (what does this mean?). If published, this will include your full peer review and any attached files.

Reviewer #1: No

---

## [Author Response · Author response to Decision Letter 0]

4 Sep 2024

Regarding the comments from specific reviewer and academic editor, we have provided the responses as Answers in the "Responses to Reviewers.pdf" file. Those are "III. Responses to Reviewer #1" in pages 8 –12, and "I. Responses to Academic Editor" in page 2, respectively.

---

## [Editor Report · Decision Letter 1]

4 Oct 2024

The N-terminal activation function AF-1 domain of ERα interacts directly with the C-terminal AF-2-holding ligand-binding domain to recruit the coactivator proteins

PONE-D-24-22407R1

Dear Dr. Liu

We’re pleased to inform you that your manuscript has been judged scientifically suitable for publication and will be formally accepted for publication once it meets all outstanding technical requirements.

Kind regards,

Jin-Ming Zhou

Academic Editor

PLOS ONE
---

## [Editor Report · Acceptance letter]

9 Oct 2024

PONE-D-24-22407R1 

PLOS ONE

Dear Dr. Liu, 

I'm pleased to inform you that your manuscript has been deemed suitable for publication in PLOS ONE. Congratulations! Your manuscript is now being handed over to our production team.

Kind regards, 

on behalf of

Dr. Jin-Ming Zhou 

Academic Editor

PLOS ONE